# Neurally Adjusted Ventilatory Assist in Acute Respiratory Failure—A Narrative Review

**DOI:** 10.3390/jcm11071863

**Published:** 2022-03-28

**Authors:** Michele Umbrello, Edoardo Antonucci, Stefano Muttini

**Affiliations:** SC Anestesia e Rianimazione II, Ospedale San Carlo Borromeo, ASST Santi Paolo e Carlo—Polo Universitario, Via Pio II, 3, 20153 Milan, Italy; edoardo.antonucci@unimi.it (E.A.); stefano.muttini@asst-santipaolocarlo.it (S.M.)

**Keywords:** acute respiratory failure, neurally adjusted ventilator assist, proportional ventilation, lung-protective ventilation, diaphragm-protective ventilation

## Abstract

Maintaining spontaneous breathing has both potentially beneficial and deleterious consequences in patients with acute respiratory failure, depending on the balance that can be obtained between the protecting and damaging effects on the lungs and the diaphragm. Neurally adjusted ventilatory assist (NAVA) is an assist mode, which supplies the respiratory system with a pressure proportional to the integral of the electrical activity of the diaphragm. This proportional mode of ventilation has the theoretical potential to deliver lung- and respiratory-muscle-protective ventilation by preserving the physiologic defense mechanisms against both lung overdistention and ventilator overassistance, as well as reducing the incidence of diaphragm disuse atrophy while maintaining patient–ventilator synchrony. This narrative review presents an overview of NAVA technology, its basic principles, the different methods to set the assist level and the findings of experimental and clinical studies which focused on lung and diaphragm protection, machine–patient interaction and preservation of breathing pattern variability. A summary of the findings of the available clinical trials which investigate the use of NAVA in acute respiratory failure will also be presented and discussed.

## 1. Introduction

Competing evidence has suggested both beneficial and deleterious consequences of spontaneous breathing during assisted ventilation when compared to controlled mechanical ventilation, in patients with acute respiratory failure [1]. In fact, maintaining spontaneous breathing has been attributed various physiological advantages: improved ventilation–perfusion matching [2], improved hemodynamics [3], reduced likelihood of ventilator-induced lung injury [4], enhanced diaphragm function with reduced muscle atrophy [5].

Furthermore, spontaneous breathing during mechanical ventilation, when improperly applied, may itself aggravate lung injury [6]. Pathways leading to such patient self-induced lung injury include regional lung stress and strain [7], patient–ventilator asynchrony [8], augmented transvascular pressure and pulmonary edema [9], diaphragmatic myotrauma [10]. Minimization of these effects by neuromuscular blockage could be one of the underlying mechanisms associated with the improved outcome when spontaneous breathing abolition is applied in the first hours of acute respiratory distress syndrome [11].

Proportional modes of ventilation, such as proportional assist ventilation with load-adjustable gain factors (PAV+) and neurally adjusted ventilatory assist (NAVA), have the potential to deliver lung and respiratory muscle protective ventilation by preserving the physiologic defense mechanisms against both lung overdistention and ventilator overassistance, as well as reducing the incidence of diaphragm disuse atrophy, while optimizing patient–ventilator synchrony [12]. In this review, we will focus on the available evidence about the potential benefits which NAVA ensures while improving the match between patients’ needs and ventilator-delivered assistance. This work briefly describes NAVA technology and clinical implementation and summarizes the clinical impact associated with lung- and diaphragm-protective ventilation, enhanced breathing pattern variability and patient–ventilator interaction, during acute respiratory failure. We will also provide a summary of the available trials which have investigated the use of NAVA in acute respiratory failure patients.

## 2. Lung and Respiratory Muscles Protective Ventilation

### 2.1. Lung Injury

Two main mechanisms contribute to the occurrence of lung injury: volutrauma or barotrauma (overdistension) and atelectrauma (reiterate collapse and recruitment of the alveoli) [13]. It is increasingly acknowledged that excess energy applied to the lung, irrespective of whether it is generated by the machine (ventilator-induced lung injury, VILI) or by the patient himself (patient self-inflicted lung injury, P-SILI) [6,14] may induce or worsen lung injury [15], possibly as a result of regional stress amplification [16]. Vigorous spontaneous efforts induce large variations in transpulmonary pressure, mainly in the dorsal regions, and air redistribution from non-dependent to dependent regions, i.e., occult pendelluft [17]. Moreover, negative pleural pressure swings generated by spontaneous inspiratory efforts can drag the alveolar pressure below PEEP. The consequent increase in transmural pulmonary vascular pressure [9] may lead to the development of VILI because of increased vascular leakage [4,18,19].

### 2.2. Patient–Ventilator Asynchronies

Patient–ventilator dyssynchrony is the uncoupling of the ventilator (mechanical) delivered breath and patient (neural) respiratory effort. Such mismatch can be classified into timing and flow assist asynchrony: the former relates to a discrepancy between the timing of the patient neural respiratory cycle and that of the ventilator and can potentially happen during triggering, insufflation and cycling off, with extremes such as auto-triggering and ineffective efforts [20]. Flow assist asynchrony refers to a discrepancy between the amplitude of the neural respiratory output and the level of inspiratory assist provided by the ventilator.

During assisted ventilation, desynchronization of patient effort and ventilator support occurs commonly and is related with ICU and hospital length of stay and duration of mechanical ventilation [21,22,23]. If untreated, it might lead to a vicious cycle which is associated with increased mortality [8,24]: occurrence of double and reverse triggering, which then may lead to increased transpulmonary pressures, greater tidal volumes, pendelluft generation and subsequent lung injury [25,26]. On the other hand, ineffective efforts are often underestimated and associated with various factors, such as excessive assist level or sedation, both resulting in compromised respiratory drive [27]. Overassistance promotes prolonged inspiratory time, late cycling, hyperinflation and intrinsic PEEP [28], which in turn increase the threshold to trigger the ventilator and therefore facilitates ineffective efforts [23,29]. In conventional assisted modes of ventilation, it is indeed possible to bring neural and mechanical inspiratory time closer by reducing pressure support and increasing the flow threshold for cycling off [30]. Nevertheless, detection and treatment of patient–ventilator asynchronies remains a complex task in the clinical practice.

### 2.3. Diaphragm Injury—Myotrauma

An inadequate titration of mechanical support can also injure the respiratory muscles, leading to myotrauma and the so-called ventilator-induced diaphragm dysfunction (VIDD) [10,31]. Both ventilator over- and underassistance have been associated with rapid alterations of diaphragm structure and function [32]: inflammation due to excessive inspiratory effort [33,34,35], as well as overassistance and respiratory drive suppression [36,37]. Moreover, inadequate PEEP levels might lead to alveolar collapse during expiration and the occurrence of eccentric myotrauma [38], as well as a shorter fiber length which is associated with less efficiency and longitudinal atrophy [39]. Diaphragm contraction during dyssincronies such as reverse triggering and ineffective efforts leads to the same pathological pathway.

Integration of lung-protective ventilation principles and the growing concept of diaphragm-protective ventilation is the key to a new approach: targeting an effort level that can protect the respiratory system from both hazards [5,40].

## 3. Proportional Ventilation

Proportional ventilatory modes are designed to optimize patient–ventilator interaction and deliver lung and respiratory-muscle protective ventilation [41]. These methods benefit patient neural control mechanisms, which are physiologically active against both lung under- and overdistension and, consequently, against diaphragm atrophy or structural damage [42]. Aiming to comply with patient ventilatory demands, respiratory assist is provided proportionally to patient effort, in terms of pressure and timing, during the whole inspiratory cycle [43].

While under mechanical assistance, both the patient and the ventilator participate to generate the pressure needed to overcome elastic and resistive forces, as outlined by the equation of motion of the respiratory system [44]:(1)$$\mathrm{Ptot}=\mathrm{Pmus}+\mathrm{Pvent}=\mathrm{PEEP}+\mathrm{Vt}\cdot \mathrm{Ers}+\;\dot{V}i\;\cdot \mathrm{Raw}$$

Compared to conventional modes of assist, proportional modes change the relationship between patient effort and tidal volume, the slope of which depicts the efficiency of the respiratory system [12]. Assuming a linear relationship between Pmus and PaCO_2_, tidal volume increases approximately in a linear fashion with inspiratory effort during unassisted ventilation [45]. During pressure support ventilation, a constant pressure is provided, causing an upward displacement of the patient effort–tidal volume relationship, without any variation in its slope [46]. As a consequence, depending on pressure support level, underassistance could occur in high-respiratory-drive patients, exposing them to P-SILI and diaphragm load-induced injury [26]. Conversely, if the patient is able to trigger the ventilator in the absence of any additional effort, a minimum volume will always be delivered, depending on pressure support level and respiratory system mechanics [47].

When the respiratory drive is fulfilled by the minimum volume generated without diaphragm engagement, a significant overassistance takes place, leading to excessive tidal volume, neural-mechanical mismatch, impairment of inspiratory muscles activity and function [48]. Sleep quality could also be compromised if ventilator overassistance leads to decline of patient respiratory effort to the limit of PaCO_2_ threshold, leading to sleep apnea events [49].

On the other hand, proportional modes increase the slope of patient effort and tidal volume function: the pressure provided by the ventilator increases proportionally with the Pmus [42]. This factor constitutes the physiological principle by which lung and diaphragm protection occurs during proportional ventilation and, as the patient themself settles the assist entity, these modes have the potential to streamline the implementation of ventilatory support [50].

Well-described physiological mechanisms occur to prevent lung overdistention under such modes: the Hering–Breuer inflation–inhibition biological feedback suppresses the respiratory drive in response to high tidal volumes [51]. Moreover, at increased lung volumes, diaphragm muscular fibers are located at an unfavorable position of the length–force relationship and respiratory system compliance decreases [52]. Otherwise, in conventional assist modes, increasing the pressure support level leads to increased tidal volume regardless of neural drive inhibition [53].

During proportional ventilation, the Pmus-Vt function starts from null values, implying a minimum respiratory muscle necessary activity, and the pressure provided is zeroed whenever the patient’s effort terminates [21]. Thereby, patient–ventilator synchrony is guaranteed during the whole respiratory cycle, whilst overassistance, patient self-induced lung injury, underassistance diaphragm atrophy and sleep apnea are far less likely to occur.

Patient–ventilator synchronization, breathing variability, neuromuscular coupling and gas exchange are improved with proportional modes: all these mechanisms potentially provide lung and diaphragmatic protective ventilation [54].

Two different proportional modes of ventilation are available in clinical practice: proportional assist ventilation with load-adjustable gain factors (PAV+) and neurally adjusted ventilatory assist (NAVA) [42,55]. Since both modes share the same operational principles, i.e., delivering inspiratory assist in proportion to the patient’s effort, they both potentially share their beneficial effects on lung and diaphragm protection as well as on patient–ventilator interaction.

PAV+ supplies a ventilatory assist proportional to the instantaneous volume and flow generated by inspiratory muscles contraction. Assessment of respiratory mechanics is achieved by using the equation of motion of the respiratory system: the machine performs automated occlusions and calculates respiratory system resistance and elastance [56,57]. A bed-side adjustable gain value then determinates the amount of force to be unloaded from patient’s respiratory effort. Triggering and cycling-off are determined with conventional techniques based on pressure or flow thresholds, similar to conventional assisted modes [43].

## 4. NAVA—Neurally Adjusted Ventilator Assist

The autonomous respiratory system integrates peripheral chemical and neural afferents at brain stem level and generates the respiratory cycle pattern. The respiratory center is part of a network regulated by complex neural feedback, and is coordinated in the pons. There are three main respiratory stimuli in healthy subjects: chemical, metabolic and voluntary components. The respiratory neural activity is implemented with signals coming from mechanoceptors located in the lungs, respiratory muscles and the chest wall. The voluntary control system is located in cortical and over-medullary structures. The phrenic nerve runs the neural signal to induce action potential in diaphragm muscle fibers, the intensity and frequency of which establish the number of motor units activated and therefore the entity of mechanical contraction. Negativization of intrathoracic pressure, induced by ribcage expansion, results in air flow during spontaneous ventilation. The integration of these processes is defined as neuroventilatory coupling, and is depicted in Figure 1.

### 4.1. Basic Principles of NAVA

NAVA is an assist mode, which supplies the respiratory system with a pressure proportional to the integral of the electrical activity of the diaphragm (EAdi) [55]. The ventilator obtains the diaphragm electromyographic signal by a nasogastric tube fitted with several (generally, eight pairs) electrodes [55,58]. Optimal NAVA catheter positioning is pivotal, and it can be achieved by applying a validated anatomical index based on the presumed distance between the crural diaphragm and the tip of the nasogastric tube, and evaluating the electrocardiographic aspect of the P and QRS waves, which is indicative of the position of the electrodes in the signal trace, and the synchrony of the diaphragm electromyographic signal with the negative deflection of the airway pressure curve during an inspiratory effort against an occluded artificial airway [59]. Possible reasons for low EAdi, despite correct esophageal tube placing, are pneumatic overassistance, excessive sedation, apnea events, phrenic nerve lesions, neuromuscular diseases and muscular weakness.

EAdi is proportional to the intensity of the electrical field produced by the contraction of diaphragm muscular fibers and, at the present time, is the closer possible measure of the activity of the respiratory centers (Figure 1). Therefore, the inspiratory peak of diaphragm electrical activity (EAdi, peak) is considered a reliable proxy of central respiratory drive in either healthy individuals or acute respiratory failure patients [52,60]. Notably, EAdi is an accurate representation of patient neural output to the extent that the diaphragm is used as the main inspiratory muscle and assuming that both phrenic nerve and the neuromuscular junction (i.e., the patient’s neuro-ventilatory coupling) are intact [48]; recent evidence proves that vagal-mediated pulmonary volume feedback is preserved even in the early phase after bilateral lung transplantation, despite surgical vagotomy distal to bronchial anastomoses [61].

The area under the inspiratory EAdi curve (EAdi, AUC) is an estimate of diaphragm’s force-generating demand and sustainability of muscular contraction over time [28,62]. The mismatch between efficiency of respiratory muscles and respiratory workload determinates neural drive and force-generating capacity disassociation: under conditions such as respiratory underassistance or weaning failure, the EAdi, peak value will increase disproportionately more than the EAdi, AUC [62].

### 4.2. NAVA Catheter Positioning

Reliable positioning of the NAVA catheter is necessary to trace a representative EAdi signal and therefore deliver a reliable input for ventilator assistance. The optimal catheter position was defined by the stability of the EAdi signal, diaphragm electrical activity highlighted in the two central ECG leads of the catheter positioning tool during inspiration and the absence of p-wave on ECG in distal lead (Figure 2) [59]. These indications are based upon investigations where an accurate diaphragmatic activity was detected as an electromyographic signal from the catheter central electrodes highest in central frequency and reduced in root mean square [63]. In fact, when the signal is derived from central leads of the electrode array it is more efficient, preventing loss of the electromyographic input caused by diaphragm displacement along the respiratory cycle.

Brender implemented the information derived from transoesophageal electrocardiogram interpretation, considering the relative position of right atrium and diaphragm [64]. In the presence of sinus rhythm, a p-wave amplitude that decreases from cranial to caudal catheter leads and its absence in the most distal lead suggests a position caudal to the right atrium. Accordingly, QRS complexes amplitude will be higher in the upper leads and decrease in the lower leads displayed.

A simplified approach is based on the formula including measurement from Nose to Ear lobe to Xiphoid process of the sternum (NEX value) [65], modified for EAdi-catheter placement (NEXmod), which proved to be accurate prediction of catheter positioning [59]. Positive end-expiratory pressure, body position and intra-abdominal pressure are known to influence the position of the diaphragm [66]; however, NAVA ventilation is not impaired when optimal catheter position is correctly ensured: the multiple-electrode distance compensates for any possible diaphragmatic positional changes [67].

Data on normal EAdi reference values are limited [68]: differences in age, anatomical characteristics, presence of an underlying chronic lung condition and variability in the distance between the catheter and the crural diaphragm were all reported to have an influence on the absolute EAdi amplitude for a given neural output [55,69]. Patient-specific EAdi thresholds have been suggested with normalization of the EAdi value to the maximal amplitude value obtained during a voluntary maximal inspiration [70]. However, translation of this principle to critically ill patients is often limited by the inability to obtain an inspiratory effort on request. Phrenic nerve electromagnetic stimulation has been used for research purposes to normalize EAdi amplitude [71], and a physiologic study showed that peak EAadi and peak EAadi/EAdimax ratio can deliver similar information [72].

### 4.3. NAVA Ventilation

Under neutrally adjusted ventilator assist, the EAdi signal, measured in microvolts, is multiplied by a user-controlled gain factor, the NAVA level (cmH_2_O/μV), so that at every time, the pressure delivered to the respiratory system is:(2)Paw=(NAVA level·EAdi)+PEEP

The timing and intensity of the EAdi signal determine the timing and intensity of the ventilatory assist, resulting in a high level of synchrony between the neural respiratory cycle and the flow of the ventilator, both in terms of time and flow assist. The pressure delivered by the ventilator is then directly proportional to both EAdi and NAVA level, and the airway pressure–time outline accurately reflects the EAdi profile (Figure 3).

Contrary to most conventional modes of assisted mechanical ventilation that use solely a pneumatic trigger, NAVA takes advantage of the EAdi signal as an “electric” trigger: the ventilator triggers when EAdi amplitude increases (usually >0.5 μV) above the baseline and cycling off depends on reduction of the signal at a fixed (preset 70%) percentage of the peak value. The use of electrical trigger and cycling-off criteria allows for preservation of the physiological variability of respiratory drive, improved patient–ventilator synchrony and unloading of respiratory muscles while avoiding overassistance. Moroever, EAdi is independent from pressure and flow generated in the respiratory system, therefore, assisted ventilation is guaranteed even in presence of high PEEP or air leaks [55].

### 4.4. Neuro-Ventilatory Efficiency Index (NVE) and Patient–Ventilator Breath Contribution (PVBC)

NVE describes the capacity of the respiratory muscles to generate pressure and ventilatory volume. It can be calculated as the ratio between tidal volume and EAdi during an unassisted respiratory act (i.e., during neurally assisted ventilation with NAVA level set at 0) [73]. This index has been proposed as a tool to evaluate patient readiness for extubation [74].

To quantify the relative contribution of the patient versus the ventilator to the inspiratory tidal volume, Grasselli and colleagues proposed the PVBC, calculated as the ratio of NVE during assisted and unassisted breaths [73].

### 4.5. Pmusc/EAdi Index or Neuro-Mechanical Efficiency Index (PEI or NME)

Bellani et al. developed an index that relates the electrical activity of the diaphragm during NAVA (EAdi) to the pressure generated by the respiratory muscles (Pmusc) [75] NME quantifies the amount of pressure the respiratory muscles can generate, normalized to EAdi, and which can be obtained at the bedside during a brief end-expiratory pause as the ratio between Paw and EAdi variations [76]. Pmusc and EAdi are related, through a proportionality coefficient, with some degree of interpatient variability, although a stability within each patient under different modes or levels of ventilator assistance was described. However, further investigation is required with NME, as repeated measurements within an individual patient exhibited unacceptably high variation and there was no correlation between NME variability and clinical parameters [76].

### 4.6. NAVA Level Setting

Several strategies have been proposed to adjust the NAVA level gain factor, each with its principles, advantages and drawbacks. The main methods are reported in Table 1.

The first method proposed is based on a “pressure matching” setting with the NAVA preview option. A curve is displayed on the ventilator in a different color, which shows an appraisal of the airway pressure that would be delivered if NAVA mode were applied. The NAVA preview (Paw) curve profile is proportional to diaphragm electrical activity and the amount of assist depends on EAdi amplitude and NAVA level setting.

Notably, the mechanical energy supplied to the lung, even when the peak pressures are similar, is lower in NAVA than PSV because proportionality implies a smaller area under the Paw curve. Hence, it is possible to set an inspiratory assist to obtain peak or mean airway pressure values similar to the ones previously reached during PSV (Paw matching). However, the application of this principle entails the risk of frustrating the full-range of potential advantages of NAVA ventilation itself [53], as the level of support is not set according to the extent of patient inspiratory drive.

Another NAVA level setting requires targeting the same minute ventilation obtained in PSV with protective volume at a fixed respiratory rate (ventilation matching) [77]. However, as previously described, during NAVA, neither tidal volume nor minute ventilation is directly under the user’s control; as a consequence, the efficiency of this method depends on the initial PSV setup, again not taking advantage of proportional ventilation benefits.

More recent methods have been developed to make the best use of EAdi potentials. NAVA level titration can be performed by systematically rising the gain factor to identify an ideal level of respiratory muscle unloading [78]. Brander and colleagues hypothesized a stepwise NAVA titration procedure starting from a low level of assistance, using a breathing pattern analysis [64]. The authors described a two-phase response consisting in a first rise in airway pressure and tidal volumes accompanied by inspiratory muscle effort (esophageal PTP) and EAdi decrease in response to a NAVA level increase (first response), followed by no variation in tidal volume and airway pressure, with esophageal PTP and EAdi reduction for further augmentation in NAVA level (second response). The transition point between the first and second response was labeled as the optimal NAVA level, identified as the threshold between initial insufficient assistance and satisfaction of patient’s neural respiratory demand. Clinical application of this approach is still controversial as clear recognition of these two phases is not obvious [79].

Another EAdi-guided method was proposed by Rozé and colleagues: the authors arbitrarily titrated NAVA level targeting an EAdi equivalent to 60% of the amplitude peak produced in a daily spontaneous breathing trial (SBT), and applied for the first time a method which was based directly on the observation of diaphragm activation [80]. Systematic use of this method has resulted in a gradual EAdi decrease until extubation. The only available clinical trial which used this titration approach showed a greater number of ventilator-free days and a shorter duration of mechanical ventilation for patients undergoing NAVA, when compared to PSV [81].

Campoccia et al. assessed the feasibility of titrating the NAVA assist on different levels of muscle unloading via the application of the neuro-ventilatory efficiency index [82]. Moderate unloading targets (40%) were feasible to obtain and were associated with greater diaphragm activity and improved ventilation homogeneity, in terms of redistribution of ventilation to the dorsal-dependent lung regions.

### 4.7. Effects of NAVA on Lung Protection

As described earlier, NAVA allows physiological ventilation protective mechanisms to operate unhampered. Since during proportional ventilation, the mechanical inflation strictly pursues patient ventilatory demand, if inspiratory time is reduced because of Hering–Brauer reflex activation, then the tidal volume delivered will be proportionally lower. Moreover, as the airway pressure-time curve has a triangular-shaped profile, yielding lower mean Paw and transpulmonary pressures compared to PSV, this may prevent VILI occurrence. NAVA has a beneficial effect on the ventilation of the dependent regions in patients with injured lungs [83]; data obtained from an animal ARDS model suggest a reduction in recruitment–derecruitment events and, therefore, atelectrauma-related VILI [84].

Brander and colleagues indicated that VILI (wet-to-dry ratio and IL-8 concentration, tissue factor plasminogen activator inhibitor II and broncho-alveolar fluid) and non-pulmonary organ damage were significantly lower in NAVA and conventional low Vt controlled ventilation strategy compared to high tidal volume-controlled ventilation in 27 rabbits with induced ARDS [85].

During assisted ventilation, a reliable measure of respiratory system mechanics is necessary to protect the patient from the risk of self-inflicted lung injury [6,86]. Grasselli and colleagues measured Pplat by means of an end-inspiratory occlusion maneuver during NAVA and observed a good correlation with values of Crs recorded in PSV. Considering that several investigators reported that reliable measurements of Pplat can be obtained during PSV [9], the authors suggest that measuring Pplat, and consequently assess respiratory mechanics, is feasible and accountable during NAVA [87]. Availability of this information, integrated with an estimation of Pmusc derived from diaphragm electrical activity, could be used to maintain protective transpulmonary pressure during proportional ventilation in NAVA [88]. Eventually, comparison between PSV and proportional modes during exercise in 10 critically ill patients suggested a beneficial effect of proportional ventilation when ventilator demands increase over time: exercise with proportional modes was associated with a better work efficiency and less increase in VO_2_ than with PSV, while the ventilator modes did not affect patients’ dyspnea, limb fatigue, distance, hemodynamics and breathing pattern, suggesting a possible enhancement of the training effect and facilitation of rehabilitation [89].

### 4.8. Effects of NAVA on Diaphragm Protection

Proportional ventilation may also facilitate a diaphragm-protective ventilation, avoiding both over- and underassistance. The effects of NAVA and PEEP on breathing pattern during experimentally induced acute lung injury have been studied in a small animal model [90]. The authors vagotomized a group of rabbits that underwent a protocol of NAVA level and PEEP titration over two periods of time and compared the findings with non-vagotomized rabbits. Vagal-mediated reflexes induced a tonic EAdi in order to keep the lung open; lung recruitment, maintained by higher PEEP levels, determined higher phasic diaphragm activity and NAVA allowed to unload the diaphragm while delivering protective tidal volumes and improved overall respiratory mechanics in the long-term groups, suggesting a potential benefit for spontaneous breathing with proportional ventilation in ARDS.

Shimatani et al. suggested that maintaining spontaneous breathing, both with NAVA and PSV, could be beneficial on preventing diaphragm atrophy [91]. The authors evaluated four groups of lung-injury-induced white rabbits (non-ventilated, CMV with neuromuscular blockade, PSV and NAVA) and reported no between-groups differences in physiological index, respiratory parameters and histologic lung injury. The NAVA group had fewer asynchrony events, a smaller fractional area of sarcomere disruption, reduced proportion of apoptotic cells and decreased expression of Caspase-3 mRNA when compared with PSV, and it was suggested as a potentially better mode to prevent asynchrony and, consequently, diaphragm sarcomere injury and apoptosis.

Recently, Scharffenberg and colleagues conducted a randomized study to assess lung and diaphragmatic function in a ARDS-induced porcine model [92]. Twenty-four pigs were randomized in three groups (NAVA, noisy PSV and PCV), reporting no differences in global alveolar damage. Gas exchange and asynchrony rate did not differ between groups. Of note, NAVA resulted in higher respiratory pattern variability and less interstitial edema in dependent lung regions when compared to noisy PSV. Lung tissue IL-8 concentrations was lower in NAVA group.

Clinical data suggest that NAVA may facilitate weaning [93]—from a pathophysiological point of view, the prevention of diaphragmatic atrophy may be one of the mechanisms underlying this evidence [94]. In a clinical trial, the electrical activity of the diaphragm during NAVA was in the same range as a previous study linked to preserved muscular thickness [95,96]. Consistently, NAVA was associated with improved diaphragm mechanical efficiency [97].

### 4.9. Effects of NAVA on Breathing Pattern Variability

NAVA ensures greater variability and therefore a more physiologic breathing pattern, which is one of the cofactors determining gas exchange improvement. EAdi reflects natural adaptation targeting the preservation of PaCO_2_ values; moreover, ventilatory rate and tidal volumes are adjusted to optimize work of breathing and lung stress and strain. The electromyographic signal of diaphragm activity integrates information provided by feedback systems from respiratory system mechanoceptors, vagal afferences and central respiratory centers [98,99]. NAVA improved the breathing pattern in terms of tidal volume and respiratory rate variability in porcine models [92]; consistently, healthy individuals adapt diaphragm activity at different NAVA levels to maintain tidal volume to settle PaCO_2_ [52,100]. Indeed, the ventilator setting during NAVA only partially determines tidal volume, which is ultimately set by the patient’s neural control, ultimately decreasing the risk of overassistance and lung injury [64].

### 4.10. Effects of NAVA on Optimization of Patient–Ventilator Interaction

NAVA has the potential benefit of optimizing neuromechanical coupling, reducing the likelihood of asynchrony during mechanical ventilation. As outlined above, proportional modes, by their very definition, supply a ventilatory assist as close as possible to the patient’s physiological needs. The use of EAdi as the criteria to trigger (even in patients with high intrinsic PEEP) and cycle-off the ventilatory support is the finest approach developed so far to achieve human–machine harmonization [101].

Beck et al. considered the different patterns of interaction and muscle unloading in response to two different modes of assist: ARDS was induced in 12 rabbits and various levels of PSV and NAVA assistance were compared on synchrony, breathing pattern variability, mechanical work of breathing (expressed as transdiaphragmatic pressure time product-PdiTP) and electrical energy consumption (as EAdi time product-EAdiTP) [102]. The incidence of asynchrony increased as PSV level increased; NAVA reduced asynchrony, specifically, ineffective triggering. Increasing NAVA level led to both PdiTP and EAdiTP reduction, without alterations in protective tidal volume. Conversely, EAdiTP, PdiTP and tidal volume increased after augmenting PSV to high levels. These results suggest that ventilator asynchrony is a cause of diaphragm electrical and mechanical load, and that NAVA could be a better option, compared to PSV, as an assist mode of ventilation in ARDS.

The same authors also assessed whether NAVA could supply ventilatory assistance employing non-invasive interface in rabbits [103]. Even if a high interface leak was detected, the results suggest good animal–ventilator interaction and unloading of respiratory muscles.

A prospective interventional study in 22 spontaneously breathing patients intubated for acute respiratory failure found that NAVA, as compared with PSV, reduced trigger delay and total asynchrony events from an average of 3 to 1 event/minute; no ineffective effort or late cycling was observed, and premature cycling was significantly reduced with NAVA [104].

As opposed to PSV, this ventilatory mode led to patient–ventilator interaction improvement in observational studies [105]. Schmidt et al. indicate that NAVA prevented overdistention and ineffective efforts enhancing the match between the patient and the ventilator in a crossover trial [100]. Similar conclusions were reported by Ferreira et al. and by Lamouret and colleagues [106,107]. Results from recent, larger clinical trials point in the same direction, confirming the physiological advantages of neurally adjusted ventilation assist on patient–ventilator interaction [81,94,95]. Even if these studies were not designed to demonstrate a superiority in terms of asynchrony occurrence, the overall use of sedative agents was reported to be consistently lower in NAVA arms, suggesting an enhanced patient–ventilator interaction.

In the only physiologic study comparing NAVA and PAV+ in adult patients, Akoumianaki et al. found that PAV+ performed better than NAVA when elastic load increased: the authors found that the linear correlation between tidal volume and the inspiratory integral of transdiaphragmatic pressure was weaker with NAVA than with PAV+ and PSV on account of a weaker inspiratory integral of the electrical activity of the diaphragm–transdiaphragmatic pressure linear correlation during NAVA. The authors questioned whether such a weak EAdi-PTPdi integral linear relationship during NAVA might limit its effectiveness to assist the inspiratory effort [108].

### 4.11. Possible Limitations of NAVA Ventilation

Neurally adjusted ventilation has not yet, to our knowledge, conclusively demonstrated improved patient-centered outcomes in any clinical trial, nor has it been extensively adopted into clinical practice. As an example, Hadfield et al. could not find a superiority of NAVA compared with PSV in terms of ICU or hospital stay, duration of MV or mortality [109].

Several technological aspects may relate to this limitation. During NAVA an EAdi increase >0.5 μV above baseline triggers inspiratory assist and cycling is determined by a fall at a pre-set percentage of its peak value. In the absence of a quality EAdi signal, assisted breaths can be initiated either by variations in Paw or flow, depending on which trigger happens first. Indeed, the electromyographic signal is independent from pressure and flow, thus, triggering is not affected by the presence of air-leaks or autoPEEP [55]. Nonetheless, high incidence of flow-triggered respiratory cycles during NAVA ventilation has been reported [108]. Difficulty in acquiring and maintaining a satisfactory EAdi signal occurred in 10 out of 36 (27.8%) NAVA participants and cross-over from NAVA to PSV was observed in a large RCT [109]. Di Mussi et al. reported NAVA failure in 7 out of 20 patients (35%) due to scarce EAdi synchrony or low EAdi amplitude, despite having obtained a reliable EAdi signal at baseline [97]. However, similar criticisms were not reported in other recent clinical trials [81,94,110]. Double triggering is related to a biphasic shape in EAdi signal, and it occurs more frequently in NAVA than in PSV [95,104,105]. However, given the proportionality of the assist level, the second cycle in NAVA often results in no flow generation and subsequently no breath stacking nor high Vt, which is the main concern related to double triggering in assist-controlled modes [111].

In the end, the exact reasons for the differences between the theoretical advantages and the practical use of NAVA remain to be investigated. Even though ‘technical issues’ was most commonly selected as the most important single disadvantage in a recent clinical survey, ‘low experience, skills or confidence’ were pointed out as a major barrier to acceptance and implementation of NAVA [112]. Contextual aspects including availability, cost, access to the technology, are crucial: NAVA mode is available only with specific oesophageal catheter on a dedicated ventilator produced by a single manufacturer. Human issues such as prevalent culture, varying levels of expertise, staff-to-patient ratios and admission rates may also play an important role as limiting factors.

### 4.12. Differences with Automated Weaning Systems

Automated (computerized) weaning from mechanical ventilation has recently been suggested as an attractive alternative to usual physician-driven weaning, for its potentially beneficial effects in terms of reduction of the duration of mechanical ventilation. Such modes were originally developed to overcome the clinicians’ underrecognition of patients’ ability to breathe without assistance, prolonging mechanical ventilation and increasing the incidence of its complications [113,114,115]. The main commercially available systems are adaptive support ventilation (ASV) on Hamilton (Hamilton Medical, Bonaduz, Switzerland), and Smartcare on Evita ventilators (Dräger, Lübeck, Germany). The use of such systems was associated with a significantly shorter time to the first successful breathing trial, and successful extubation, with fewer tracheostomies and episodes of extended ventilation [116]. However, the results were obtained from a highly selected population of critically ill patients, and the main limitation of such systems is that they primarily rely on assessing respiratory mechanics, patterns of breathing, and gas exchange (each measure of which has questionable reliability when assessed over brief periods) and none are currently integrated with actual measures of respiratory drive or effort, nor do they deliver proportional assist. In fact, both Smartcare and ASV use “conventional” pneumatic signals such as flow and volume [117]. NAVA, on the other hand, has the unique feature to control ventilator functioning through the electrical activity of the diaphragm (EAdi), since the mechanical support is on-triggered and cycled off by the EAdi, and is proportional to that signal throughout each inspiration [55].

## 5. Clinical Use of NAVA in Acute Respiratory Failure

Optimizing ventilator machine–patient interaction, minimizing ventilator-related and self-inflicted lung injury, reducing myotrauma are the main strategies to improve the outcome in patients with acute respiratory failure. Most clinical trials advocating the use of NAVA in patients with acute respiratory failure considered pressure support ventilation as the comparator, established treatment mode [118]. In general, the common limitation to most of the studies is that, for the sake of easier comparison between the two modes of assistance, NAVA level has generally been titrated with “conventional” pressure- or volume-based criteria (such as a similar peak airway pressure as during pressure-, or a matched respiratory rate and tidal volume). The lack of reliance on titration protocols based on each patient’s central drive might in part explain the negative results of many comparisons, as the full-range of advantages of proportional support in the NAVA group might not have been achieved due to the design of the studies. Table 2 summarizes the characteristics of the studies included in the current systematic review.

Colombo and colleagues performed the first study of NAVA in critically ill patients, evaluating the response to different levels of ventilatory assistance [119]. They included patients receiving invasive mechanical ventilation for a variable time interval and with various underlying diseases. NAVA level was titrated on Paw with NAVA preview function, while maintaining a constant sedation. The authors showed significant differences at higher assist levels in breathing pattern and respiratory drive; although Paw peak values were not different between PSV and NAVA, the former was associated with a greater ventilator assistance and a smaller neural drive. NAVA did not increase the risk of dynamic hyperinflation, while PSV demonstrated significantly prolonged mechanical insufflation exceeding the neural inspiratory time. As opposed to PSV, NAVA enhanced tidal volume variability, ensuring a more physiologic breathing pattern. All these changes have proven to be more evident at higher assist levels, suggesting that NAVA potentially limited the risk of overassistance, limited patient–ventilator asynchrony and optimized the interaction with the machine.

Demoule et al. conducted a randomized multicenter study which enrolled patients who received mechanical ventilation for more than 24 h for acute respiratory failure with a respiratory cause [94]. Sixty-two patients receiving NAVA were compared with sixty-six patients in PSV. NAVA level setting was targeted on a resulting Vt of 6–8 mL/kg IBW; both groups had the same weaning protocol, consisting of daily spontaneous breathing trials. Notably, the centers included were already experienced with NAVA mode and EAdi monitoring was available for both groups. This study was the first to demonstrate that NAVA is safe and feasible for several days, in routine critical ill patient care. NAVA enhanced patient–ventilator synchrony and resulted in lower dyspnoea occurrence and less need for post-extubation NIV. Notwithstanding the theoretical advantages, NAVA did not improve the likelihood of remaining in an assisted mode, and it did not reduce overall ICU mortality, nor the duration of mechanical ventilation. The authors suggested that NAVA could be mostly beneficial in selected patients, with specific causal factors determining respiratory failure, such as COPD, major patient–ventilator asynchrony and those difficult to wean.

Ferreira and colleagues designed a crossover trial on 20 mechanically ventilated patients who underwent an SBT in PSV or NAVA [107]. This is the first trial evaluating NAVA application continuously until extubation. The most common causes of respiratory failure were chronic obstructive pulmonary disease (COPD) exacerbation and pneumonia. NAVA level was titrated to generate an equivalent peak airway pressure, with a PEEP of 5 cmH_2_O. When compared to PSV, NAVA reduced patient–ventilator asynchrony index, by reducing triggering and cycling delay, and generated, during an SBT, a similar breathing pattern.

Liu et al. performed a randomized monocentric trial to investigate whether NAVA was more effective in difficult-to-wean patients when compared to PSV [95]. Ninety-nine patients who had already failed a first SBT or undergone reintubation were enrolled. NAVA titration was set on protective volume ventilation, with a fixed trigger of EAdi. Of note, the EAdi signal was not available for PSV patients, allowing a real comparison of pressure support ventilation as used in clinical practice. This was the first work to demonstrate that NAVA decreased the duration of weaning, increased ventilator-free days and the probability of successful weaning as compared to PSV.

Hadfield and colleagues compared NAVA and PSV in patients at risk of prolonged mechanical ventilation (i.e., those with COPD, heart failure or ARDS) in a feasibility RCT [109]. NAVA level titration was based on matching pressure delivery through NAVA preview mode. Seventy-two patients were included in four academic centers; the results showed feasibility and safety over a prolonged period of time (beyond 48 h) of NAVA application, without any adverse event. Exploratory clinical outcomes highlighted advantages of NAVA over PSV in this specific population: increased ventilator-free days, reduced time to breathing without assistance, reduced time to ICU discharge, improved sedation management and reduced hospital mortality.

Diniz-Silva and colleagues compared NAVA and PSV in providing protective ventilation in ARDS patients [111]. ARDS patients have been included in other studies, with unspecified timing or during weaning phase [21,94,120]; this crossover, single-center, randomized trial enrolled 20 patients just after neuromuscular blockage and deep sedation discontinuation. The study period was short and 25% of patients interrupted the protocol for excessive respiratory drive. The authors concluded that most patients with ARDS, under continuous sedation, could be ventilated in NAVA within protective levels; NAVA and PSV resulted in similar breathing patterns, whereas NAVA resulted in a greater, although still protective, Paw than PSV. There was no difference in asynchrony index between the two groups.

Kacmarek et al. carried out a multicenter, randomized controlled trial under the hypothesis that NAVA, compared to conventional lung-protective mechanical ventilation, may determine benefits on ventilator-free days and mortality in patients with acute respiratory failure [81]. The study enrolled 306 patients with hypoxemic or hypercapnic ARF, ventilated for less than 5 days and expected to require ventilation for >72 h; patients with moderate–severe ARDS were excluded. At variance with all the other studies, NAVA was used throughout the entire course of patients’ need for ventilatory support; NAVA level titration was based on achieving a EAdi at about 50% of the maximum EAdi peak obtained during a short time without ventilation assistance. The authors found that NAVA increased the number of ventilator-free days, shortened the duration of ventilation in ICU survivors and reduced reintubation rate, when compared to conventional modes.

Some recent systematic reviews and meta-analyses compared either NAVA alone or proportional modes (NAVA and PAV+) with conventional assist ventilation: Yuan et al. included 7 studies suggesting that NAVA might be superior to PSV in difficult-to-wean patients [93]; Chen et al. found a lower asynchrony index with NAVA vs. PSV and no significant differences in respiratory muscle unloading, with NAVA being associated with a significantly shorter duration of ventilation despite a similar ICU length of stay or mortality [121]; Kataoka et al. found that the use of proportional modes was associated with a reduction in the incidence of asynchronies, weaning failure and duration of mechanical ventilation, compared with PSV; however, reduced weaning failure and duration of mechanical ventilation were found with only PAV and not NAVA [54].

## 6. Conclusions

Neurally adjusted ventilation assist provides the potential for lung and diaphragm protective ventilation. This ventilatory mode allows physiological mechanisms to minimize the probability of volutrauma and atelectrauma, patient–ventilator asynchrony and myotrauma, while optimizing breathing pattern variability and patient–machine interaction. Indeed, the same beneficial effects of NAVA on lung and diaphragm protection and patient–ventilator interaction might apply to PAV+ as well, since both proportional modes share the same operational principles.

Setting NAVA level may be challenging in everyday clinical practice: inspiratory assist, respiratory muscle effort and unloading all need to be tailored to each patient and clinical situation. During NAVA, an optimal inspiratory assist level titration can be achieved by different means. Different approaches, based on various physiological assumptions, have been explored so far and this represents an interesting field for further investigation.

In conclusion, various clinical trials concluded that NAVA mitigate the risk of overassistance, reduced patient–ventilator asynchrony and improved patient–ventilator interaction, leading to a reduced duration of mechanical ventilation. These results seem to be even more promising in specific conditions and setting. While we wait for more conclusive evidence regarding the impact of neurally adjusted ventilatory assist on patient-centred outcomes, we suggest that the application of NAVA in acute respiratory failure patients might lead to clinically relevant results, especially if the whole range of advantages of proportional ventilation are considered with a careful titration of the inspiratory assist.

## Figures and Tables

**Figure 1 jcm-11-01863-f001:**
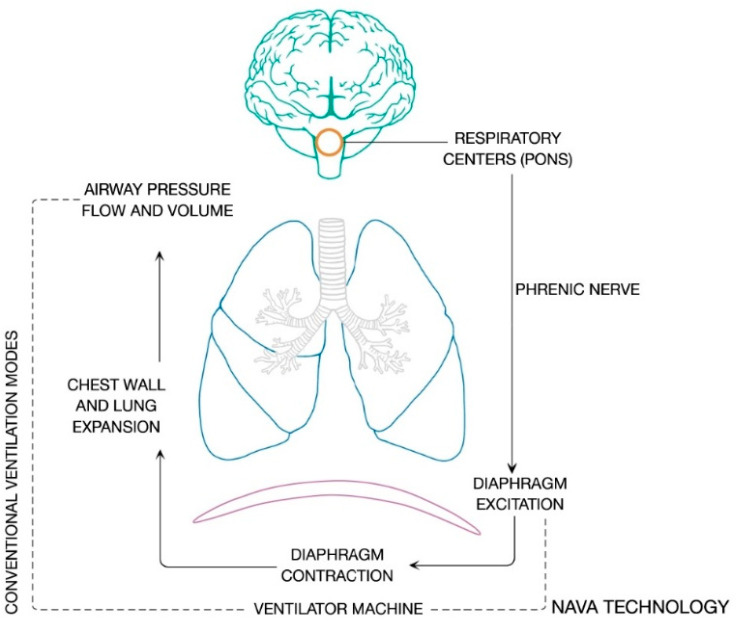
Neuroventilatory coupling process. The figure shows the integration of peripheral chemical and neural afferents at the respiratory centers and the efferent signals which are generated under the neuroventilatory coupling.

**Figure 2 jcm-11-01863-f002:**
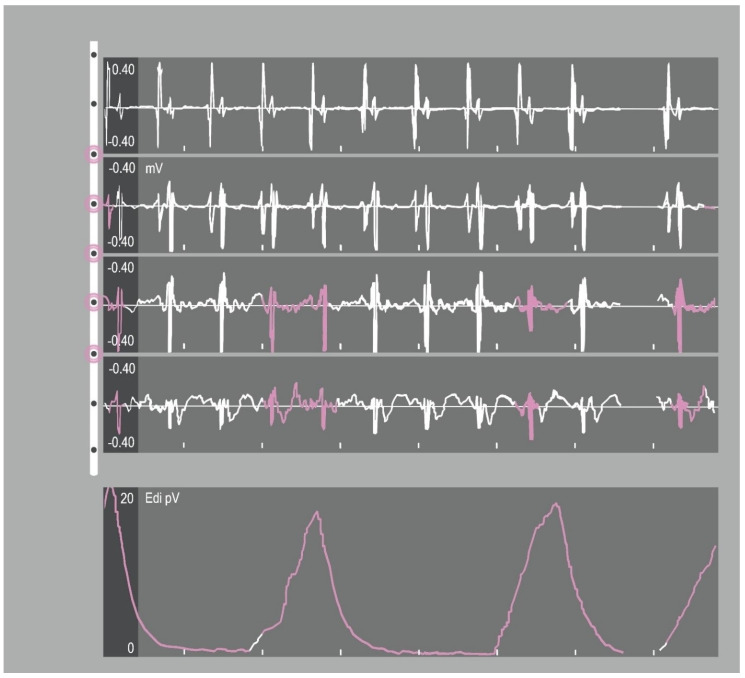
Catheter-positioning tool screen. The tool shows the progression of the retrocardiac electrocardiogram (ECG) complexes. The electrical activity of the diaphragm (Edi) signal (bottom tracing) is superimposed over the EKG tracings and should be in the middle tracings for optimal placement.

**Figure 3 jcm-11-01863-f003:**
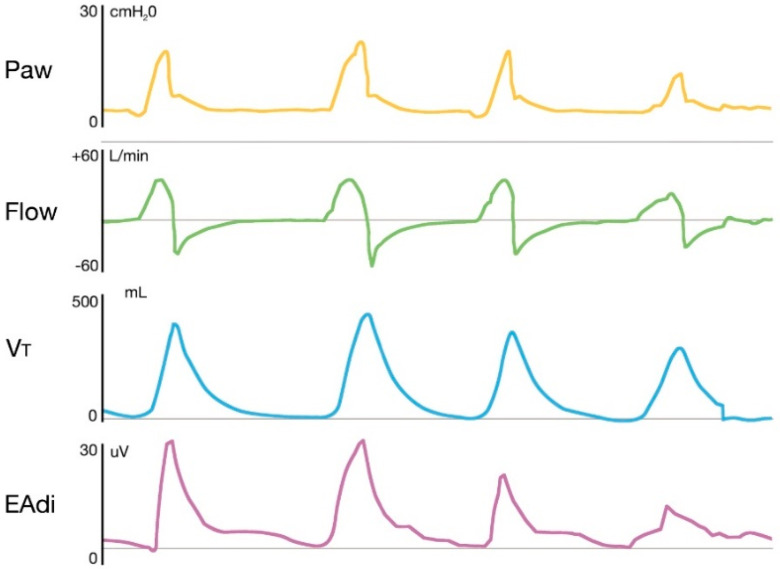
Representative tracings of airway pressure, flow, tidal volume and electrical activity of the diaphragm in a patient receiving NAVA. The figure shows how the pressure delivered by the ventilator is directly proportional to both EAdi and NAVA level, and the airway pressure-time outline accurately reflects the EAdi profile. Paw: Airway pressure; Vt: Tidal volume; EAdi: electrical activity of the diaphragm.

**Table 1 jcm-11-01863-t001:** NAVA level setting methods.

Method		Advantages	Disvantages	Reference
Conventional approach. (does not integrate EAdi signal)	Mean or peak airway pressure matching (NAVA preview)	Easy to use. Possible use as a monitoring tool to detect asynchronies method in PSV mode.	Does not consider variation in EAdi caused by PSV to NAVA transition. Breathing pattern variability in EAdi may determinate difficult comparison. Paw peak matching does not guarantee similar assist levels due to differences in pressure curve shape. Depends on initial PSV titration quality.	Cecchini et al., 2014
	Ventilation matching	Easy to use.	Tidal ventilation in NAVA is not under the user’s control. Depends on initial PSV titration quality.	Coisel et al., 2010
Patient’s response-based approach	Biphasic breathing pattern response	Physiological method. Reflects patient’s muscular activity. Proved to result in a more personalized assistance level compared to NAVA preview methods.	Not obvious recognition of transition point (curvilinear relationship between EAdi and Pmusc), e.g., high-respiratory-drive patients.	Brander et al., 2009
	Percentage of EAdi peak during SBT	Physiological method. Direct observation of diaphragm activity. Provides periodical reassessment of the NAVA level and EAdi	Limited to use after a negative SBT. Maximum EAdi during SBT may be different according to the SBT setting and method. Does not consider accessory respiratory muscles. It may result in deleterious high inspiratory efforts in patients with high respiratory drive.	Rozè et al., 2011
	Ventilatory muscles unloading (NVE based)	Physiological method. Easy to perform at the bedside. Recommended to use moderate unloading target	Limited to the weaning phase. NVE does not directly represent breathing effort.	Campoccia et al., 2018

**Table 2 jcm-11-01863-t002:** Characteristics of the studies included in the systematic review.

Author, Year	Study Type	Etiology and Inclusion	Sample Size	Design	Intervention	Control	Conclusions
Colombo et al., 2008	Crossover, prospective, randomized, controlled trial	All intubated patients receiving partial ventilatory support	14	Physiological, 20 min duration	NAVA Paw peak-titrated support level to PSV	PSV Support level set to obtain protective tidal volume	NAVA mitigated the risk of overassistance, reduced patient–ventilator asynchrony, and improved patient–ventilator interaction.
Demoule et al., 2016	Parallel, multicenter, randomized trial	De novo hypoxemic respiratory failure, acute on chronic respiratory failure, acute cardiogenic pulmonary edema; Patients on MV > 24 h for ARF	128	Clinical, weaning phase (14 days); weaning failure defined as the need for switching to a controlled mode	NAVA Ventilation-titrated support level	PSV Support level set to obtain protective tidal volume, PEEP set according to local guidelines	NAVA is safe and feasible; it does not increase the probability of remaining in assisted ventilatory mode. NAVA decreases patient–ventilator asynchrony and is associated with less frequent application of post-extubation NIV.
Ferreira et al., 2017	Randomized, monocentric crossover trial	COPD, pneumonia, pleural effusion, sepsis, coma, trauma, drowning, cardiac failure, cardiac arrest; Patients on MV > 48 h and considered ready for SBT	20	Physiological, 30 min SBT duration	NAVA Airway peak pressure matching	PSV Support level 5 cmH_2_O, PEEP level 5 cmH_2_O	NAVA reduces patient–ventilator asynchrony and generates a respiratory pattern similar to PSV during SBTs. Safe submission to SBT in NAVA.
Liu et al., 2020	Randomized, monocentric clinical trial	COPD, pneumonia, sepsis, acute cardiogenic shock, neurologic disease, surgery; Difficult-to-wean patients; Invasive MV > 24 h	99	Clinical, difficult weaning patients	NAVA Ventilation-titrated support level	PSV No EADi signal available Support level set to obtain protective tidal volume PEEP set to maintain SpO_2_ >90%	In patients difficult to wean, NAVA decreased the duration of weaning and increased ventilator-free days.
Hadfield et al., 2020	Open-label, parallel, multicenter, randomized controlled trial	COPD, heart failure, ARDS; Patients at risk of prolonged MV	72	Feasibility in weaning phase, mode adherence and protocol compliance beyond 48 h	NAVA Paw titrated-support level EAdi target 8 µV	PSV Support level set to obtain protective tidal volume	Good adherence to assigned ventilation mode and ability to meet a priori protocol criteria. Exploratory outcomes suggest clinical benefit for NAVA compared to PSV.
Diniz-Silva et al., 2020	Prospective, monocentric, randomized, crossover trial	Pneumonia, aspiration, anaphylactic shock ARDS, MV > 24 h, inspiratory efforts for more than 6 h	20	Feasibility, provide protective ventilation in ARDS patients	NAVA Airway peak pressure matching	PSV support level set to generate tidal volume < 6 mL/kg PBW	NAVA is feasible as a protective ventilation strategy in selected ARDS patients, under continuous sedation
Kackmarek et al., 2020	Multicenter, randomized, controlled trial	ARF patients (heterogeneous etiologies); MV < 5 days	306	Clinical, patients expected to require MV ≥ 72 h	NAVA Level titration: EAdi 50% of the maximum EAdi peak obtained during an SBT	PSV Support level set to obtain protective tidal volume	NAVA decreased duration of MV, it did not improve survival in ventilated patients with ARF.

## Data Availability

Not applicable.

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
