# Peer review of "Neurally Adjusted Ventilatory Assist in Acute Respiratory Failure—A Narrative Review"

_jcm, 2022, doi:10.3390/jcm11071863_

Round 1
Reviewer 1 Report
Well written review, nice and useful figures/tables, updated with the last references.
I would only suggest the Authors to cite this "Rauseo, Michela, and Lise Piquilloud. "Proportional modes." Heunks L and Schultz M. Handbook Invasive mechanical ventilation. The European Respiratory Society, (2019): 62-73.", for sake of completeness.
Author Response
Well written review, nice and useful figures/tables, updated with the last references.
I would only suggest the Authors to cite this "Rauseo, Michela, and Lise Piquilloud. "Proportional modes." Heunks L and Schultz M. Handbook Invasive mechanical ventilation. The European Respiratory Society, (2019): 62-73.", for sake of completeness.
R: thanks for the comments. The suggested reference was added to the reference list.
Reviewer 2 Report
I have read with interest the narrative review on NAVA. The review provides updated information and is well written. Nevertheless, I have some concerns regarding this work.
- My main concern is the structure of this review. In general is too long and somewhat heterogenous. It starts with extensive analysis of lung and diaphragm protective ventilation giving the impression that the main purpose is to analyze lung and diaphragm protection and not NAVA. NAVA is introduced on page 5! On the other hand, there is no information on how to insert the EAdi catheter, find correct position, recognize and address problems with EAdi signal, how flow trigger is activate or that NAVA automatically changes to PSV when EAdi is unreliable. A whole paragraph is dedicated comparing NAVA with automated modes while there is no paragraph analyzing NAVA insertion, problems with NAVA, limitations etc. With respect to the structure, lung and diaphragm protection should be incorporated in a paragraph analyzing the advantages of NAVA compared to PSV.
- In line with my previous comment, a paragraph commenting on the limitations of NAVA should be provided: NAVA implementation requires an esophageal catheter and a dedicated ventilator. Problems with EAdi signal are common. As an example, difficulties in acquiring and, more importantly, maintaining a satisfactory EAdi signal occurred in 10 out of 36 (27.8%) NAVA participants in the study of Hadfield. Di Mussi et al. reported NAVA mode failure in seven out of 20 patients (35%) due to loss of ‘Edi synchrony’ or low Edi activity, despite having obtained a reliable Edi signal at baseline. Piquilloud L et al. and Akoumianaki E. et al in their physiologic studies reported that 30% of ventilator cycles were flow triggered during NAVA due to problems with EAdi start. Double triggering is a special problem with NAVA resulting from biphasic EAdi signal.
- The review does not report the results of recent systematic reviews and meta-analysis comparing either NAVA alone or proportional modes (NAVA and PAV) with conventional assist ventilation:
- Yuan X. et al, Meta-Analysis Crit Care 2021 Jun 29;25(1):222. doi: 10.1186/s13054-021-03644-z: a meta-analysis of 7 studies suggesting that NAVA might be superior than PSV in difficult to wean patients.
- Chen C. et al. Ann Transl Med 2019 Aug;7(16):382. doi: 10.21037/atm.2019.07.60: A meta-analysis comparing NAVA with PSV in patient-ventilator interaction and clinical outcomes: lower AI with NAVA, no significant differences in respiratory muscle unloading (EAdipeak, P 0.1, VT/EAdi). For clinical outcomes, NAVA was significantly lower than the PSV (MD -2.82, 95% CI: -5.55 to -0.08, I2=0%) in the duration of ventilation, but two groups did not show significant differences in ICU mortality, ICU stay time, and hospital stay time.
- Kataoka J, Kuriyama A, Norisue Y, Fujitani S. Proportional modes versus pressure support ventilation: a systematic review and meta ‑ Ann Intensive Care. doi:10.1186/s13613-018-0470-y. Meta-analysis comparing PSV vs proportional ventilation (PAV-NAVA). The use of proportional modes was associated with a reduction in the incidence with AI > 10%, weaning failure and duration of mechanical ventilation, compared with PSV. However, reduced weaning failure and duration of mechanical ventilation were found with only PAV.
- In a physiologic study, Akoumianaki E et al., compared PSV, PAV+ and NAVA. This is the only study comparing NAVA and PAV+ in adult patients and exhibited interesting results. PAV+ performed better than NAVA when elastic load increased. The authors found that the linear correlation between tidal volume and inspiratory integral of transdiaphragmatic pressure was weaker with NAVA than with PAV+ and PSV on account of a weaker inspiratory integral of the electrical activity of the diaphragm (∫EAdi)-PTPdi linear correlation during NAVA. Interestingly, the weak ∫EAdi-PTPdi linear relationship during NAVA might limit its effectiveness to assist the inspiratory effort. This study merits attention.
- Comment that no study found a benefit of NAVA on survival. As an example the study of Hadfield et al. (2020) did not find a superiority of NAVA compared with PSV in terms of ICU or hospital stay, duration of MV or mortality.
- Page 8, lines 274-280. It should be emphasized that the neuromuscular efficiency index requires further investigation. Repeated measurements of NMEoccl within an individual patient exhibited unacceptably high variation and there was no correlation between NMEoccl variability and clinical parameters as showed by Jansen D. et al. (refer 65).
- The study of Piquilloud et al. addressing patient-ventilator synchrony with NAVA should be cited in the paragraph analyzing patient-ventilator interaction with NAVA (Piquilloud LV, et al. Neurally adjusted ventilatory assist improves patient-ventilator interaction. Intensive Care Med. 2011;37(2):263–71.)
- Comparison between PSV and proportional modes during exercise indicates a beneficial effect of proportional ventilation when ventilator demands increase over time (less increase in oxygen consumption) and this could be added in the advantages of NAVA (Can proportional ventilation modes facilitate exercise in critically ill patients? A physiological cross-over study : Pressure support versus proportional ventilation during lower limb exercise in ventilated critically ill patients. Akoumianaki E, et al. Ann Intensive Care. 2017 Dec;7(1):64. doi: 10.1186/s13613-017-0289-y.
- To avoid repetition and to provide a more smooth reading, I would propose to delete the paragraph ‘NAVA in Acute Respiratory Failure’ and analyze the effects of NAVA in patients with ARF as (1) effects on breathing variability (2) effects on patient-ventilator synchrony (3) effects on lung protection (4) effects on diaphragm protection (5) effects on outcomes (weaning, survival, ICU stay, extubation etc. Table 2 is useful to summarize the results of the main studies.
Author Response
I have read with interest the narrative review on NAVA. The review provides updated information and is well written. Nevertheless, I have some concerns regarding this work.
- My main concern is the structure of this review. In general is too long and somewhat heterogenous. It starts with extensive analysis of lung and diaphragm protective ventilation giving the impression that the main purpose is to analyze lung and diaphragm protection and not NAVA. NAVA is introduced on page 5! On the other hand, there is no information on how to insert the EAdi catheter, find correct position, recognize and address problems with EAdi signal, how flow trigger is activate or that NAVA automatically changes to PSV when EAdi is unreliable. A whole paragraph is dedicated comparing NAVA with automated modes while there is no paragraph analyzing NAVA insertion, problems with NAVA, limitations etc. With respect to the structure, lung and diaphragm protection should be incorporated in a paragraph analyzing the advantages of NAVA compared to PSV.
R: thanks for the comments, which helped us to improve the clarity and flow of our manuscript. As suggested, the first part was significantly shortened, and two new paragraphs were added: “NAVA catheter positioning” (which includes a new explicative figure – now figure 2) and “Possible limitations of NAVA ventilation”. Nevertheless, we still think that a brief introduction (significantly shortened by at least 40% of its length as compared to the previous version) on lung and diaphragm protective ventilation is useful to put NAVA into context, as we believe that this proportional mode is able to deliver such a ventilation. We are however available to move this paragraph if the Reviewer or the Editor think it should be done.
- In line with my previous comment, a paragraph commenting on the limitations of NAVA should be provided: NAVA implementation requires an esophageal catheter and a dedicated ventilator. Problems with EAdi signal are common. As an example, difficulties in acquiring and, more importantly, maintaining a satisfactory EAdi signal occurred in 10 out of 36 (27.8%) NAVA participants in the study of Hadfield. Di Mussi et al. reported NAVA mode failure in seven out of 20 patients (35%) due to loss of ‘Edi synchrony’ or low Edi activity, despite having obtained a reliable Edi signal at baseline. Piquilloud L et al. and Akoumianaki E. et al in their physiologic studies reported that 30% of ventilator cycles were flow triggered during NAVA due to problems with EAdi start. Double triggering is a special problem with NAVA resulting from biphasic EAdi signal.
R: as stated responding to comment 1, a new paragraph named “Possible limitations of NAVA ventilation” has been added, as suggested. The paragraph includes the comments and quotes the papers suggested by the reviewer
- The review does not report the results of recent systematic reviews and meta-analysis comparing either NAVA alone or proportional modes (NAVA and PAV) with conventional assist ventilation:
- Yuan X. et al, Meta-Analysis Crit Care 2021 Jun 29;25(1):222. doi: 10.1186/s13054-021-03644-z: a meta-analysis of 7 studies suggesting that NAVA might be superior than PSV in difficult to wean patients.
- Chen C. et al. Ann Transl Med 2019 Aug;7(16):382. doi: 10.21037/atm.2019.07.60: A meta-analysis comparing NAVA with PSV in patient-ventilator interaction and clinical outcomes: lower AI with NAVA, no significant differences in respiratory muscle unloading (EAdipeak, P 0.1, VT/EAdi). For clinical outcomes, NAVA was significantly lower than the PSV (MD -2.82, 95% CI: -5.55 to -0.08, I2=0%) in the duration of ventilation, but two groups did not show significant differences in ICU mortality, ICU stay time, and hospital stay time.
- Kataoka J, Kuriyama A, Norisue Y, Fujitani S. Proportional modes versus pressure support ventilation: a systematic review and meta ‑ Ann Intensive Care. doi:10.1186/s13613-018-0470-y. Meta-analysis comparing PSV vs proportional ventilation (PAV-NAVA). The use of proportional modes was associated with a reduction in the incidence with AI > 10%, weaning failure and duration of mechanical ventilation, compared with PSV. However, reduced weaning failure and duration of mechanical ventilation were found with only PAV.
R: a paragraph was added to discuss the findings of these 3 systematic reviews and meta-analysis, as suggested
- In a physiologic study, Akoumianaki E et al., compared PSV, PAV+ and NAVA. This is the only study comparing NAVA and PAV+ in adult patients and exhibited interesting results. PAV+ performed better than NAVA when elastic load increased. The authors found that the linear correlation between tidal volume and inspiratory integral of transdiaphragmatic pressure was weaker with NAVA than with PAV+ and PSV on account of a weaker inspiratory integral of the electrical activity of the diaphragm (∫EAdi)-PTPdi linear correlation during NAVA. Interestingly, the weak ∫EAdi-PTPdi linear relationship during NAVA might limit its effectiveness to assist the inspiratory effort. This study merits attention.
R: this interesting study was discussed, as suggested
- Comment that no study found a benefit of NAVA on survival. As an example the study of Hadfield et al. (2020) did not find a superiority of NAVA compared with PSV in terms of ICU or hospital stay, duration of MV or mortality.
R: A reference to this issue and the study by Haddad was added, as suggested
- Page 8, lines 274-280. It should be emphasized that the neuromuscular efficiency index requires further investigation. Repeated measurements of NMEoccl within an individual patient exhibited unacceptably high variation and there was no correlation between NMEoccl variability and clinical parameters as showed by Jansen D. et al. (refer 65).
R: A sentence was added to express this concern, as suggested
- The study of Piquilloud et al. addressing patient-ventilator synchrony with NAVA should be cited in the paragraph analyzing patient-ventilator interaction with NAVA (Piquilloud LV, et al. Neurally adjusted ventilatory assist improves patient-ventilator interaction. Intensive Care Med. 2011;37(2):263–71.)
R: this interesting study was discussed, as suggested
- Comparison between PSV and proportional modes during exercise indicates a beneficial effect of proportional ventilation when ventilator demands increase over time (less increase in oxygen consumption) and this could be added in the advantages of NAVA (Can proportional ventilation modes facilitate exercise in critically ill patients? A physiological cross-over study : Pressure support versus proportional ventilation during lower limb exercise in ventilated critically ill patients. Akoumianaki E, et al. Ann Intensive Care. 2017 Dec;7(1):64. doi: 10.1186/s13613-017-0289-y.
R: this interesting study was discussed, as suggested
- To avoid repetition and to provide a more smooth reading, I would propose to delete the paragraph ‘NAVA in Acute Respiratory Failure’ and analyze the effects of NAVA in patients with ARF as (1) effects on breathing variability (2) effects on patient-ventilator synchrony (3) effects on lung protection (4) effects on diaphragm protection (5) effects on outcomes (weaning, survival, ICU stay, extubation etc. Table 2 is useful to summarize the results of the main studies.
R: we fully understand the change proposed by this reviewer. However, we respectfully think that the current structure reflects the fact that the majority of the studies summarized in the sections “Effects of NAVA on lung protection”, “Effects of NAVA on diaphragm protection”, “Effects of NAVA on breathing pattern variability”, “Effects of NAVA on optimization of patient-ventilator interaction” were preclinical or physiological studies or were performed in patients who were recovering from acute respiratory failure; with the “NAVA in Acute Respiratory Failure” paragraph we aimed to summarize the evidence on the use of NAVA in patients during the acute phase of respiratory failure. To make this clearer, we changed the title of this section to “CLINICAL USE OF NAVA IN ACUTE RESPIRATORY FAILURE”.
Round 2
Reviewer 2 Report
I congratulate the authors for the revised version. The structure has significantly improved and references are now updated. I would suggest, as a minor comment, to add a few sentences indicating that the beneficial effects of NAVA on lung and dipahragm protection and patient/ventilator interaction apply to PAV+ as well simce proportional modes share the same operational principles.